# FTA Cards as a Rapid Tool for Collection and Transport of Infective Samples: Experience with Foot-and-Mouth Disease Virus in Libya

**DOI:** 10.3390/ani12223198

**Published:** 2022-11-18

**Authors:** Fadila Abosrer, Giulia Pezzoni, Emiliana Brocchi, Anna Castelli, Stefano Baselli, Santina Grazioli, Hafsa Madani, Elfurgani Kraim, Abdunaser Dayhum, Ibrahim Eldaghayes

**Affiliations:** 1National Center for Animal Health, Tripoli P.O. Box 83252, Libya; 2Department of Microbiology and Parasitology, Faculty of Veterinary Medicine, University of Tripoli, Tripoli P.O. Box 13662, Libya; 3Istituto Zooprofilattico Sperimentale della Lombardia e dell’Emilia Romagna (IZSLER), 25124 Brescia, Italy; 4Institut National de la Médecine Vétérinaire, El Harrach, Alger P.O. Box 205, Algeria; 5Department of Preventive Medicine, Faculty of Veterinary Medicine, University of Tripoli, Tripoli P.O. Box 13662, Libya

**Keywords:** FTA cards, foot-and-mouth disease, RT-PCR, field, sequence

## Abstract

**Simple Summary:**

Foot-and-mouth disease (FMD) is a highly contagious viral disease affecting cloven-hoofed domestic and wild animals. FMD can lead to high economic losses for farmers due to a decrease in animal production and high mortality in young animals. The rapid and accurate diagnosis of an FMD outbreak is the first and very important step in order to control and prevent the further spread of the FMD virus. There is no cross-protection between different FMD serotypes; hence, having specific vaccines to combat the circulating FMD serotypes is crucial to control the disease. Many countries, especially developing countries, face difficulties in diagnosing FMD and sending samples to reference laboratories for diagnostic confirmation and virus characterization. Using Flinders Technology Associates (FTA) cards can be a cheap alternative method for sending clinical samples at room temperature with an ordinary courier.

**Abstract:**

Foot-and-mouth disease (FMD) is a viral disease, widespread and highly contagious, that mainly affects cloven-hoofed domestic and wild animals. FMD can lead to high economic losses due to the reduction in animal production such as a drop in milk production, loss of body weight, and a high mortality rate in young ruminants. Sixteen samples were collected from animals showing typical clinical signs of FMD during the last FMD outbreak in Libya in 2018–2019. Flinders Technology Associates (FTA) cards impressed with blood, swabs, or vesicular epithelium samples were shipped to the WOAH FMD reference laboratory in Brescia, Italy, and tested for the detection of FMD viruses. Nucleic acids were extracted from the FTA cards, and molecular testing based on real-time RT-PCR assays was carried out, of which one was specifically designed for the detection of the FMD virus of serotype O, topotype O/East Africa-3 (O/EA-3), that was further confirmed by a sequence analysis of the VP1 gene. The phylogenetic analysis of the VP1 gene showed a nucleotide identity of more than 99% between the virus circulating in Libya and the FMD virus strains isolated in Algeria in 2019.

## 1. Introduction

Foot-and-mouth disease (FMD) is a widespread and highly contagious transboundary viral disease that can be counted as one of the most economically important and devastating diseases of livestock due to the diversity of the affected animal species, the rapid spread between and within geographic regions, and its control difficulties. It affects almost all wild and domesticated cloven-hoofed mammals [1].

The disease is characterized by vesicular lesions and erosions of the epithelium of the mouth, nose, muzzle, feet, teats, and udder. Fever (up to 42 °C), acute depression, and a decrease in milk production are the most common clinical signs. The tongue, hard palate, gums, lips, udder, teats, vulva, prepuce, and interdigital membrane of the foot develop blister-like lesions or vesicles. Within 24 h, the vesicles rupture, leaving raw, painful sores surrounded by ragged tags of necrotic epithelium [2]. Excess salivation, lip-smacking, and eating cessation are all symptoms of the painful stomatitis linked with the unruptured and freshly ruptured vesicles in extreme cases. There is a rapid decrease in physical conditions. Foot lesions are accompanied by acute lameness and a reluctance to walk. Secondary mastitis might make teat lesions even more problematic. The lesions are usually less noticeable in sheep and goats [3,4].

The FMD virus is a nonenveloped virus with a positive-sense, single-stranded RNA genome. It belongs to the genus *Aphthovirus* of the viral family *Picornaviridae* [1]. The FMD virus has seven serotypes (O, A, C, Asia 1, Southern African Territories (SAT) 1, SAT2, and SAT3) that are immunologically distinct with no cross-protection between them [5]. In addition, each FMD virus serotype has a number of genetic variants called topotypes [6].

The FMD virus has four structural proteins: VP1, VP2, VP3, and VP4. VP1, a highly variable protein, has an important role in the virus’ attachment to and entry into the host cell. It is also involved in protective immunity by inducing neutralizing antibodies, and it contributes to serotype specificity. The sequencing of VP1 provides a practical way to distinguish between different FMD virus topotypes and lineages; in fact, the phylogenetic analysis of FMD viruses is largely based on VP1 sequencing. For serotype O, which is the most widespread, 11 topotypes are known: Euro-SA for the Europe-South America circulating strains; ME-SA for Middle East-South Asia; SEA for the Southeast Asia topotype; CHY for Cathay; WA for West Africa; EA-1, EA-2, EA-3, and EA-4 for East Africa; and ISA-1 and ISA-2 for Indonesia-2 [7,8]. Seven of these eleven FMDV serotype O topotypes are cocirculating in the African continent [9].

The control of FMD is mainly based on the use of vaccines; however, due to the high immunological variability, vaccination may fail to protect against antigenically different subtypes within the same serotype. Thus, there is a necessity for the rapid and accurate identification of the circulating FMD virus to evaluate the best vaccine to be used [10].

There are many obstacles and difficulties in Libya regarding shipping samples to international reference laboratories. Hence, using Flinders Technology Associates (FTA) cards could be an alternative, practical way to store and send clinical samples. Indeed, the FTA cards were created for the easy and safe transport and storage of biological samples for molecular diagnostics. FTA cards are cotton-based cellulose cards with chemicals that burst cells, denature proteins, and preserve nucleic acids for a long time at room temperature, which results in a sample suitable for molecular identification and lowers the risk of disease transmission. FTA card technology has been evaluated for many viruses as an effective and safe option for transporting hazardous samples from sampling sites to laboratories at room temperature [11,12,13,14,15]. The usefulness of FTA cards has also been demonstrated for the collection, shipment, storage, and identification of FMDV genomes in previous experimental studies [16,17], which in addition showed the virus spotted on the FTA^®^ cards could not be isolated, confirming effective inactivation of viral infectivity. However, another specific study showed that efficient rescue of live FMD virus can be achieved by chemical transfection of RNA extracted from FTA cards [11].

In this communication, we report on the use of FTA cards as a unique method available in Libya during the FMD epidemic 2018–2019 to ship samples collected from animals showing clinical signs of FMD to a reference laboratory overseas for laboratory diagnostic confirmation and further genotyping analyses.

## 2. Materials and Methods

### 2.1. Samples

Sixteen samples including swabs, epithelial tissues, and blood samples were collected from animals with clinical FMD symptoms (10 cattle and 6 sheep). Samples were collected from three different cities in western Libya, i.e., Tripoli, Tajoura, and Misrata (Table 1).

Swab samples: Swabs were sampled from the oral cavity of six animals (one cattle and five sheep) by swabbing the oral mucosa and tongue and putting the swabs into 2 mL of sterile phosphate-buffered saline (PBS). The swabs were immediately kept on ice and then stored at −80 °C for further processing.

Epithelial tissue: The affected epithelial tissue was sampled from the mouth and tongue of seven cattle and one sheep. At least one gram of the epithelium was placed in a transfer medium of PBS or equal parts of a glycerol-phosphate solution with a pH of 7.2–7.6, and all samples were immediately kept on ice and then stored at −80 °C for further processing.

Blood samples: A minimum 10 mL of EDTA-stabilized blood sample was collected from two of the cattle, placed on ice and then stored at −80 °C for further processing.

### 2.2. FTA Cards

Due to the lack of viral isolation facilities and because there are no reagents available in the Tripoli laboratory to perform RT-PCR for FMDV diagnosis, the samples were stored in the deep freezer while arranging shipment to an international reference laboratory for diagnosis. However, because of the instability of the country, and lack of available couriers to safely send clinical samples, the use of FTA cards was proposed as an alternative method to send samples abroad. FTA cards (Whatman plc, Little Chalfont, Buckinghamshire, UK) were labeled and coded for each of the above-mentioned samples. Samples were taken out of the deep freezer, and impressions of the epithelial tissues, swabs, and drops of blood samples on FTA cards were left to dry and then prepared to be shipped by a currier under a UN3373 category B parcel to the WOAH FMD reference laboratory (Istituto Zooprofilattico Sperimentale della Lombardia e dell’Emilia Romagna (IZSLER)) in Brescia, Italy.

### 2.3. RNA Extraction

The method used to extract RNA from the FTA cards consisted of a simple elution according to Muthukrishnan et al. [16,17], briefly: six sections of 5 mm in diameter were removed from each FTA card using a disposable biopsy punch (Kai medical industries, Seki-shi, Japan). The six paper disks were divided into two 1.5 mL test tubes (three in each tube), 500 µL of minimum essential medium (MEM) was added, and the tubes were incubated overnight at +4 °C. For each sample the RNA was extracted from 140 µL of the overnight elution in MEM from one of the two duplicate tubes by using the QIAamp viral RNA extraction mini Kit (Qiagen, Hilden, Germany).

### 2.4. Real-Time RT-PCR

Two real-time RT-PCR tests were performed on the extracted RNA: the first one, targeting the 3D gene of FMDV genome, is able to detect all FMD virus serotypes [18]. The second was a serotype specific assay targeted to VP1, contextually designed by IZSLER to detect isolates belonging to FMD virus O/EA-3 topotype, which was circulating in the nearby regions of North Africa. Primers and probe for the serotype-specific real-time RT-PCR were designed on the sequence of an O/EA-3 isolate from Algeria 2018 (O/ALG/19Z000124/2018 GenBank accession number ON939565.1.) and are shown in Table 2. To evaluate the absence of PCR inhibitors, an exogenous control was included in the 3D reaction mastermix, which consisted of a synthetic RNA produced and used as described by Vandemeulebroucke et al. [19] and Vandenbussche et al. [20]. Both reactions were conducted using the SuperScript™ III Platinum™ One-Step qRT-PCR Kit (Thermo Fisher Scientific, Waltham, MA, USA). The amplification profile was as follows: 60 °C (30 min, reverse transcription (RT) step) for 1 cycle, 95 °C (10 min, denaturation step) for 1 cycle, 95 °C (15 s, denaturation step) and 60 °C (1 min, elongation and fluorescence acquisition step) both repeated for 50 cycles, the reaction conditions were previously described by Shaw et al. [21].

### 2.5. VP1 Sequencing

Seven samples which produced positive real-time RT-PCR reactions were submitted to FMDV VP1 sequencing process. The VP1 (636 bp) amplification was performed using the primers and the conditions described by Knowles et al. [22]. However, to improve the reaction sensitivity, a semi-nested reaction was carried out: the first RT-PCR reaction was performed by using the One-Step RT-PCR kit (Qiagen, Hilden, Germany) with the following amplification profile: 50 °C (30 min, RT step) for 1 cycle, 95 °C (15 min, denaturation step) for 1 cycle, 95 °C (1 min, denaturation step) for 35 cycles, 55 °C (1 min, annealing step) for 35 cycles, 72 °C (2 min, elongation step) for 35 cycles, and 5 min 72 °C for 1 cycle.

The second reaction (semi-nested) was conducted using 5 µL of the first amplification product, using the same reverse primers of the first reaction and the primer O-1C583F (GACATGTCCTCCTGCATCTG) as forward primer. The reaction conditions and amplification profile were the same as the first reaction except for RT step which was not included in the amplification profile.

The amplified product (25 µL) was loaded on 2% agarose gel and, after the migration, the band corresponding to the expected molecular weight was sliced off and purified using the kit NucleoSpin^®^ Gel and PCR Clean-up (Macherey-Nagel sourced by Carlo Erba Reagents S.r.l., Milan, Italy).

The VP1 sequencing was carried out by the Sanger method using the instrument 3500XL genetic analyzer (Applied Biosystem, Thermo Fisher Scientific, Waltham, MA, USA) and assembled in contig using the Lasergene Sequencing Analysis software package (DNAStar, Madison, WI, USA). The resulting sequences were compared to others in GenBank (using BLAST) and then aligned by Clustal W using the software MEGA 11 (Molecular Evolutionary Genetics Analysis version 11) [23]. Prototype sequences for each serotype O topotype, obtained from the WRL for FMD (https://www.wrlfmd.org/fmdv-genome/fmd-prototype-strains, accessed on 6 October 2022), were also included in the sequences dataset. Phylogenetic analysis was performed by IQ-Tree webserver (http://iqtree.cibiv.univie.ac.at/, accessed on 6 October 2022) and the parameters as Maximum Likelihood method [24,25], bootstrap analysis of 1000 replicates, and midpoint root were set. Additionally, the best fit model GTR + F + I + G4 identified by Model Finder was applied [26].

The VP1 sequence obtained from one Libyan sample (O/LIB/Misrata/2019) has been submitted to GenBank with the following accession number ON256571.

## 3. Results

### 3.1. Detection of Serotype O of FMD Virus on FTA Cards

Seven samples out of sixteen eluted from impression smears on FTA cards tested positive for both the pan-FMDV real-time RT-PCR targeted on the 3D gene and the experimental real-time RT-PCR designed on the VP1 of the O/EA-3 topotype (Table 3); all seven positive samples were bovine epithelial tissues impressions. In contrast, swabs and blood samples tested negative. The absence of PCR inhibitors in these negative samples was proven by the amplification of the exogenous control included in the 3D reaction. FMDV genome was not detected in any sample that originated from sheep.

Six out of the seven positive samples were from Misrata and the seventh was from Tajoura, and all of them were collected in 2019. Samples collected in 2018 from Tripoli, including only sheep, were negative for FMD.

### 3.2. Confirmation of the Topotype by VP1 Sequencing

All seven samples, where the presence of FMDV genomes was demonstrated by the positivity of two real-time RT-PCR reactions, were submitted to the VP1 amplification protocol for sequencing. For sample number three (Table 4), the amplification occurred by means of the semi-nested protocol. The analysis of the VP1 sequence demonstrated that the virus belongs to topotype O/EA-3 with nucleotides identity ranging from 99.7–99.8% with the Algerian topotype O/EA-3 isolated during the FMD outbreak in Algeria in 2019 (Table 4). The phylogenetic analysis confirms the Libyan strain to cluster within isolates from the FMD 2018–2019 epidemic wave in the Maghreb (Figure 1).

## 4. Discussion

The FMD situation varies across North African countries including Libya. Historically, four FMDV serotypes have been reported in North Africa (O, A, SAT2, and C) with serotype O being the most prevalent serotype, followed by serotype A [27].

In the current study, we have described the diagnostic pathway to confirm an FMD outbreak in Libya in 2019 and perform phylogenetic analyses by using FTA cards for collection and safe transport of suspect samples.

Nucleic acids can be preserved on FTA cards and stored at room temperature and shipped without the need to maintain the cold chain. At the laboratory, RNA of the FMD virus can be extracted from the FTA card, and molecular diagnostic techniques can be performed to reach the final diagnosis [16,17,28].

In the present study, 16 biological samples were collected from animals showing clinical signs of FMD and FTA cards were used as a unique means that was available in Libya at that time to send dangerous samples to an international reference laboratory. The extracted RNA was analyzed by two real-time RT-PCRs, namely one based on 3D genes, which is able to detect all FMDV strains, and one specifically designed to detect the serotype O topotype East Africa 3 (EA3), which was the topotype circulating in Maghreb in 2018–2019. While the 3D real-time RT-PCR is a well-standardized test that is extensively validated and routinely used by all international reference laboratories [29], the VP1 real-time RT-PCR was developed as a prototype diagnostic test for the specific detection of the FMD virus supposed to be present in those FTA transported samples; despite its experimental characteristic, it reproduced the same qualitative results as the validated 3D-targeted diagnostic test.

In particular, seven cattle samples, corresponding to FTA cards smeared with vesicular epithelial tissues collected in 2019, scored positive with both tests, thus confirming the circulation in Libya of FMD virus serotype O belonging to topotype EA-3. In contrast, the FMDV genome was not detected from FTA cards spotted with oral swabs or blood. The positive/negative test results strongly depend on the original virus concentration in the different samples; indeed, the vesicular epithelium is indicated as the optimal sample for FMD diagnostic tests as the viral loads is normally higher in the epithelial tissues compared to swabs and blood [30]. Furthermore, the viremic phase during FMDV infection occurs before the appearance of vesicles, thus in animals with clinical signs the presence of the virus in the blood is not likely to happen. Interestingly, samples that originated from sheep and collected one year earlier (2018) all scored negative; the disease is often subclinical in sheep or causes only mild clinical signs and this could explain why only oral swabs were available from sheep. This also supports the assumption that virus load in the sheep samples was below the detectable level of the test. However, the hypothesis that FMD did not circulate in Libya in 2018 and those samples were true negative cannot be excluded.

Previous studies conducted by Muthukrishnan et al. [16] showed that the comparison of Ct values obtained in real-time RT-PCR from FMD viral RNA extracted from field samples fixed and unfixed onto FTA resulted in a loss of a range of 2–22 Ct values for FTA samples. Thus, if a sample has a poor viral load, it is likely not to be detected by using FTA cards for storage; accordingly, it would be preferable to use the FTA cards to transport high viral payload samples, such as from the vesicular epithelium, to succeed in the detection and sequencing. Indeed, it was possible to sequence the entire VP1 from a single sample out of the seven FMDV-positive samples, namely the one for which the real-time RT-PCR reaction targeted for VP1 occurred with a lower Ct value (sample # 3, Ct 22.13). The difficulty we actually encountered in amplifying the whole VP1 gene for sequencing may suggest that these samples have undergone partial genome degradation. This, however, may not affect the amplification of small genome fragments as it occurs in real-time RT-PCR reactions.

The phylogenetic analysis of the VP1 sequence obtained confirmed that the FMD virus detected in this study was serotype O topotype EA-3. The virus was closely related with an identity of more than 99% with the strains circulating in Algeria in 2019. This is a new incursion since the previous circulating strains in Libya belonged to the ME-SA topotype, which was prevalent in Libya from 2013 to 2014 [22,31]. The observed strain was primarily found in Sub-Saharan Africa; nevertheless, strains of this topotype were previously reported in many African countries [32].

Libya’s unique geographical location in North Africa makes it vulnerable to FMDV strains that can be further spread to other countries. This was the case when the O/ME-SA/Ind-2001d sublineage was first reported in Libya in 2013. It then spread to other countries in North Africa, moving from Libya to Tunisia and subsequently to Algeria in 2014 and to Morocco in 2015 [33,34]. Furthermore, previous researchers have emphasized the potential of FMDV strains prevalent in Sub-Saharan Africa to spread to North Africa [35,36].

Historically Libya encountered numerous FMDV incursions. Overall, three different serotypes have circulated: O, A, and SAT2. Type O is the most prevalent serotype, followed by serotype A [25]. In 2012 an FMDV O/EA-3 topotype was reported (FMD WRL country reports) co-circulating with ME-SA topotype PanAsia-2 lineage ANT-10 sublineage; however, those EA-3 strains were more closely related with eastern Africa strains (i.e., Ethiopia and Sudan) while the 2019 incursion was more correlated with strains circulating in the Maghreb from a western Africa route.

Illegal movement of livestock from Sub-Saharan countries creates a warning alarm for the continuous introduction of transboundary animal diseases including FMD viruses [25]. Despite many efforts by the Libyan veterinary authority to control FMD, the shortage of proper quarantining facilities and the illegal movement of animals from the surrounding countries puts Libya at high risk of continuous disease introduction.

The availability of sequence data from our region for field FMDV isolates, including sequences for FMDV collected by using FTA cards, will enable us to understand the genetic relationships between FMDV strains circulating in Libya and neighboring countries. In addition, these data can be used to rapidly trace the source of any FMD outbreak for epidemiological surveillance. One example is the introduction of a new FMDV serotype O/ME-SA/Ind-2001 that was first reported in North Africa in Libya in 2013 [37]. Moreover, the results of this work were crucial for the Libyan National Center for Animal Health, as the selection of FMDV strains that were included in the FMD vaccines were chosen based on the results obtained out of this work. Mass vaccination against FMD for cattle twice per year has been implemented in the following years; therefore, no new FMD outbreaks have been reported up to now.

## 5. Conclusions

Despite the limited number of samples submitted from Libya to a reference laboratory for FMD suspect confirmation using FTA cards, the results of testing confirmed the previous experimental evidence showing the usefulness of FTA cards to preserve and send biological samples for molecular diagnosis and genotyping analyses when conditions for regular and safe shipment of dangerous samples are not feasible. Vesicular epithelium was confirmed as the best suited sample for transport by FTA cards for FMD diagnosis due to the presence of a high viral load in the epithelium tissue. FMDV serotype O, topotype EA-3, was identified as the virus circulating in Libya in 2019.

## Figures and Tables

**Figure 1 animals-12-03198-f001:**
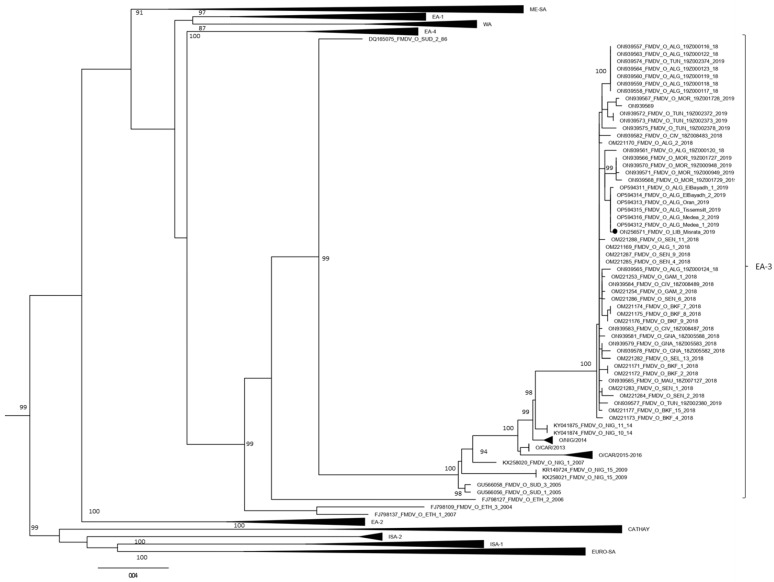
Maximum Likelihood phylogenetic tree showing the grouping of sequences from the 2018–2019 outbreak in the Maghreb within the O/EA-3 topotype. The sequence generated in this study is indicated with a black dot. Nodes bootstrap values major of 80 were indicated. The other topotypes within FMDV type O sequences are indicated. Sequences from Nigeria (NIG/2014) include GenBank accession numbers KY041874 to KY041876 and KY065150 to KY065155, and sequences from Cameroon (CAR/2015–2016) include GenBank accession numbers MG873208 to MG873223.

**Table 1 animals-12-03198-t001:** Details of samples collected on FTA cards.

Sample	Host	Lesion Age	Type of Sample	City	FMD Vaccinated	Temp.	Age	Collection Date
1	Cattle	3 Days	Epithelial tissue	Misrata	No	39.5	3 Y	May 2019
2	Cattle	2 Days	Epithelial tissue	Misrata	No	39.5	3 Y	May 2019
3	Cattle	2 Days	Epithelial tissue	Misrata	No	39.5	3 Y	May 2019
4	Cattle	3 Days	Epithelial tissue	Misrata	No	N/A	N/A	May 2019
5	Cattle	2 Days	Epithelial tissue	Misrata	No	N/A	N/A	May 2019
6	Cattle	2 Days	Epithelial tissue	Misrata	No	N/A	N/A	May 2019
7	Cattle	2 Days	Oral swab	Misrata	No	39.5	3 Y	May 2019
8	Cattle	2 Days	Whole blood	Misrata	No	N/A	N/A	May 2019
9	Cattle	3 Days	Epithelial tissue	Tajoura	No	N/A	2 Y	May 2019
10	Cattle	3 Days	Whole blood	Tajoura	No	N/A	N/A	May 2019
11	Sheep	N/A	Oral swab	Tajoura	No	N/A	N/A	October 2018
12	Sheep	N/A	Oral swab	Tajoura	No	N/A	N/A	October 2018
13	Sheep	N/A	Oral swab	Tripoli	No	N/A	N/A	October 2018
14	Sheep	N/A	Oral swab	Tripoli	No	N/A	N/A	October 2018
15	Sheep	N/A	Oral swab	Tripoli	No	N/A	N/A	October 2018
16	Sheep	N/A	Epithelial tissue	Tripoli	No	N/A	N/A	October 2018

N/A: Not available.

**Table 2 animals-12-03198-t002:** Primers and probe for the real-time RT-PCR designed for the detection of topotype O/EA3 from Algeria (GenBank accession number ON939565.1).

Oligo Name	Sequence 5′ to 3′	Nucleotide Position
O_EA3_ALG_F	CTTCTTTCAACTACGGTG	482–499
O_EA3_ALG_R	GCCACTATCTTCTGTTT	604–620
O_EA3_ALG_P	FAM-CTGCTGGCAATTCACCCG-BHQ1	571–588

**Table 3 animals-12-03198-t003:** Results of real-time RT-PCR with 3D and VP1 genes target.

Sample	Host	Type of Sample	City	Real-Time RT-PCR 3D (Ct)	Real-Time RT-PCR Designed on VP1 O/EA-3 Algerian Isolates (Ct)
1	Cattle	Epithelial tissue	Misrata	30.09	31.47
2	Cattle	Epithelial tissue	Misrata	22.79	28.24
3	Cattle	Epithelial tissue	Misrata	20.23	22.13
4	Cattle	Epithelial tissue	Misrata	24.52	26.07
5	Cattle	Epithelial tissue	Misrata	21.07	28.23
6	Cattle	Epithelial tissue	Misrata	31.44	30.64
7	Cattle	Oral swab	Misrata	Undetected	Undetected
8	Cattle	Whole blood	Misrata	Undetected	Undetected
9	Cattle	Epithelial tissue	Tajoura	26.28	28.64
10	Cattle	Whole blood	Tajoura	Undetected	Undetected
11	Sheep	Oral swab	Tajoura	Undetected	Undetected
12	Sheep	Oral swab	Tajoura	Undetected	Undetected
13	Sheep	Oral swab	Tripoli	Undetected	Undetected
14	Sheep	Oral swab	Tripoli	Undetected	Undetected
15	Sheep	Oral swab	Tripoli	Undetected	Undetected
16	Sheep	Epithelial tissue	Tripoli	Undetected	Undetected

**Table 4 animals-12-03198-t004:** Most closely related sequences of Algerian FMD isolates to the Libyan isolate.

GenBank Accession Number	Virus Name	Host	Identity %	Serotype	Topotype
OP594312	O/ALG/Medea/1/2019	Cattle	99.7	O	EA-3
OP594316	O/ALG/Medea/2/2019	Cattle	99.8	O	EA-3
OP594314	O/ALG/ElBayadh/2/2019	Cattle	99.8	O	EA-3
OP594313	O/ALG/Oran/2019	Sheep	99.8	O	EA-3
OP594315	O/ALG/Tissemsilt/2019	Sheep	99.8	O	EA-3
OP594311	O/ALG/ElBayadh/1/2019	Cattle	99.7	O	EA-3

## Data Availability

Not applicable.

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
