# Peer review of "FTA Cards as a Rapid Tool for Collection and Transport of Infective Samples: Experience with Foot-and-Mouth Disease Virus in Libya"

_animals, 2022, doi:10.3390/ani12223198_

Round 1

Reviewer 1 Report

Overall, a very interesting and important manuscript. 

I would suggest only minor spelling edits. Such as: 

Line 17-18 of the abstract, does not read clearly. 

Author Response

I’d like to thank all reviewers for their exhaustive revision, valuable, generous and constructive comments and suggestions, that for sure have great impact to improve our submitted manuscript.

Just to bring the attention to the reviewers and editors that this work has not been done in normal country, to work in a country like Libya, unstable country with ongoing war and conflict for few years, is not an easy task.  

I hope that all corrections made will satisfy the reviewers and Editors expectations and to have our manuscript published in the esteemed journal “Animals”!

Reviewer 1:

The authors are examining the use of FTA cards as a tool to be used in the field for storing/shipping samples to be used in rapid detection of FMDV. Specifically, the authors have examined samples collected from symptomatic animals in the 2018-2019 FMDV outbreak in Libya and assessed if they could store and ship samples on FTA card and detect FMDV via qRT-PCR.

I consider this topic original and relevant to the field. Developing countries all over the world, especially those that experience endemic FMDV face difficulty collecting and processing animal samples, an essential step in the process of controlling the FMDV outbreak. Many countries do not have the finical or logistical means to collect and send animal samples to a reference laboratory while maintaining the sample integrity to ensure an accurate diagnostic report.

This manuscript adds information about the use of a well-known diagnostic tool for the collection of FMDV samples in the field, where other common collection methods may not be feasible.

Regarding the methodology, it would be beneficial to add the details on the limits of detection within the qRT-PCR assay by sample. Often, different sample types have different limits of detection. Have the authors determined this with all the samples have indicated in this study? This would be of particular importance since the authors found that most of the collected Oral swabs were negative for FMDV, even though they were collected from obviously symptomatic animals.

Answer: regarding the performance of the two molecular tests carried out in the present study, it has to be considered that one test (pan-FMDV) consists of the reaction based on the 3D gene which is described in the WOAH Terrestrial Manual, and it is the most widely used and extensively validated diagnostic molecular test for FMDV detection. On the other hand, the newly designed FMDV O EA/3 topotype-specific real-time RT-PCR, targeted on the VP1 structural protein-coding gene, was developed as a prototype diagnostic test, for the genomic characterization of the FMD virus supposed to be present in Libyan samples; despite its experimental character, it showed the same qualitative results as the validated 3D-targeted diagnostic test, and similar Ct values were obtained in most cases.

We think that the negative results obtained mostly depend on the original viral payload in the different samples; the vesicular epithelium is known to be the optimal sample in terms of virus concentration.

We have integrated the discussion and have commented our results in relation to this information; moreover, we have quoted results of a previous experimental study by Indian authors regarding the reduced performance in terms of Ct values observed when testing FMDV positive field samples fixed onto FTA cards compared to unfixed ones, which can further explain our results.

Additionally, the authors should include some control samples and some FMDV “spiked” samples as a control to demonstrate accuracy with known values.

Answer: we think that the answer above also covers this comment.

In the conclusion part, the authors did address the main question. However, I find there to be concern that several of the samples they collected from obviously symptomatic animals were negative by qRT-PCR. This is a concern if this method were to be used in the field to detect FMDV. This would lead to a false negative result, which could have serious consequences in an FMDV outbreak. This also suggests that may be some breakdown or degradation of the nucleic acid collected on the FTA chips. This also points out the need and potential benefit of adding additional control to this study.

Answer: to address the referee’s concern, we have integrated the discussion with the explanation that the low concentration of virus loaded on FTA cards when using oral swabs or blood resulted in insufficient amounts of targets to provide positive results. We have also commented that negative outputs were mostly from sheep which barely show clinical signs in most cases (this could explain why only oral swabs were available for sheep) and this further supports our explanation of virus amounts below the test detectable level. Concerning the possibility of nucleic acid degradation, the difficulty we actually encountered in amplifying the whole VP1 gene for sequencing may suggest partial genome degradation, which however may not affect the amplification of small genome fragments as it occurs in real-time RT-PCR.

Also, I find the reference indicated here to be appropriate.

In addition, I find that Figure 2 does not add any scientific information or value to the article. It is simply screenshots of a qRT-PCR curve with no indication of which samples correspond to which curve in the image.

Answer: we agree and thank you for this suggestion, we have deleted the figure and modified the manuscript accordingly.

Many times, qRT-PCR viremia data is presented as Log10 Genome copy number/mL. Perhaps the authors could consider calculating this and presenting this information from the Ct values.

This could provide valuable information about the primers and qRT-PCR assay used (eg insights into limits of detection of the assay especially when there are negative qRT-PCR results for symptomatic animals).

Answer: the real-time RT-PCR based on the 3D gene is described in the WOAH manual and has been extensively validated by the WOAH reference laboratories with detection limits in the range of 10-1 TCID50/mL, while the VP1-specific test is a prototype. Usually, a Genome copy number/mL quantification is not performed for FMDV, thus we are not able, given the short time due to answering, to do this experimentation.

Reviewer 2 Report

The authors report on using FTA cards for molecular detection and characterization of FMDV in Libya.

However, the manuscript seems rather weak to be presented as an article since it reports the results based on only 16 samples. It is more like a report on diagnostics applied in that particulate case; the scientific contribution is lagging. The comparisons of results based on fresh samples and FTA cards would be beneficial. The real-time RT-PCRs would be better presented and more informative if MIQE guidelines were followed and quantification data were provided. Have the authors tried the chemical transfection of extracted RNA in adequate cell line? These data would add on the quality of the manuscript.

There are some, small-to-moderate, English language corrections to be made throughout the manuscript.

Title

Lines 2-3. The title is misleading. The FTA cards are a tool for rapid field sampling. Detection and serotyping in this article took place in a laboratory. Please provide alternative title.

Introduction

Lines 25-28. The sampling is not clearly described in the Abstract section. Please rewrite.

Line 70. “…avoiding disease contamination.” Did you mean avoiding accidental disease transmission?

Lines 64-77. Please describe the background of the previous usage of FTA cards in FMDV related diagnostics. The general description of their characteristics for infectious diseases should be replaced by more focused data on FTA cards related to samples originated from FMDV infected animals and different sample types.

Materials and methods

Section 2.1-2.3 please provide the actual number of each sample type collected and refer to the table 1 for further description.

It is not clear whether these 16 samples originate from 16 animals or different sample types were collected from some individuals? Please provide clear description.

Table 1. What does “-“ mean?

Figure 1. Since FTA cards have been in use for some time, this figure seems unnecessary, or, you can provide it in the supplementary material.

Line 127. Reaction conditions were not provided. If citation Callahan et al. covers it, please rewrite the sentence in lines 126-128.

Line 128. Please explain the annealing/elongation temperature of 50.3°C.

Line 129. Primers not primes. Please correct. Add “-“ between real and time (the same error needs to be corrected throughout the manuscript) . What did you mean by profile?

Table 2. Please provide positions of designed oligos in relation to the reference FMDV genome and which target gene fragment they amplify.

Please remove Table 3 and provide short description of reaction conditions within the text or provide citation only. Furthermore, the citation you have provided (Knowles et al., 2014) is not related to primers described in Table 3. Have you used the Knowles et al. from 2016 (VP1 sequencing protocol for foot and mouth disease virus molecular epidemiology)?

Line 142. Semi-nested, please correct.

Lines 151-153. The authors have not provided a necessary information regarding the multiple sequence alignment (method) and the phylogenetic analysis (methods, models, bootstrap) they implemented.

Line 153. If you provide the information that the Pirbright Institute is WRL, then provide the information that Anses/Sciensano are the EURL for FMD.

It is not clear whether these sequences were deposited in the GenBank; accession numbers were not provided.

Results

Line 159. Please correct the typos.

Table 4. Please add an additional column for VP1 RT-PCR results.

Lines 165-166. Figure 2 does not provide VP1 sequencing information.

Figure 2 is not informative and should be removed.

Line 170. The sentence is repeated since almost the same one was written in the line 156.

Lines 172-174. To claim these samples were negative, the results of IPC amplification should be provided. No information regarding IPC was provided in materials and methods. Please introduce these results and related information on materials and methods used.

Lines 176-181. Have you used the FMDV Toolbox the additionally check the VP1 sequences?

Figure 3. The resolution is poor, please provide better Figure. Moreover, the accession numbers of all taxa in the phylogenetic tree should be provided within the tree, or as a supplementary material. Some taxa have asterisks in their name, please provide the explanation on the meaning. The overall number of taxa could be reduced in order to improve on clarity of that tree.

Discussion

Lines 206-208. The explanation should be expanded on the comparisons with other studies using FTA cards related to FMDV detection. What is the limit of detection regarding FMDV load on FTA cards?

Author Response

I’d like to thank all reviewers for their exhaustive revision, valuable, generous and constructive comments and suggestions, that for sure have great impact to improve our submitted manuscript.

Just to bring the attention to the reviewers and editors that this work has not been done in normal country, to work in a country like Libya, unstable country with ongoing war and conflict for few years, is not an easy task.  

I hope that all corrections made will satisfy the reviewers and Editors expectations and to have our manuscript published in the esteemed journal “Animals”!

The authors report on using FTA cards for molecular detection and characterization of FMDV in Libya.

However, the manuscript seems rather weak to be presented as an article since it reports the results based on only 16 samples. It is more like a report on diagnostics applied in that particulate case; the scientific contribution is lagging.

Answer: we agree that a specific study to quantitatively analyze the recovery of the FMDV RNA and its conservation status using FTA cards was not conducted; however, such studies have been experimentally fulfilled and previously reported by other authors, while the main objective of our report was to describe the experience of using FTA cards in a real case. This enabled us to conclude that, despite the limited number of samples submitted for diagnostic confirmation of FMD, the results supported previous experimental evidence of the usefulness of FTA cards to preserve and send biological samples for molecular diagnosis and genotyping analyses, when conditions for regular and safe shipment of dangerous samples are not feasible and helped to identify the best suited biological samples for FTA cards. It is also well documented that during FMD outbreak it is enough to collect samples from 5 animals per herd.  

The comparisons of results based on fresh samples and FTA cards would be beneficial. The real-time RT-PCRs would be better presented and more informative if MIQE guidelines were followed and quantification data were provided. Have the authors tried the chemical transfection of extracted RNA in an adequate cell line? These data would add on the quality of the manuscript.

Answer: other authors compared the results in terms of Ct values between fresh samples and the same samples on FTA cards, finding that FTA fixed samples resulted in a loss of up to 22 Ct values in some cases; we have reported this finding in the discussion and have commented it in relation to our observations. Moreover, we integrated the text with more information regarding the performance of the 3D real-time RT-PCR which is a well-standardized test, validated according to WAHO guidelines and routinely used by all international reference labs. The chemical transfection of RNA extracted from FTA cards is the subject of a comprehensive study previously described by Indian authors, who succeeded in live virus recovery and reported a better performance compared to conventional cell culture isolation. In our case, however, the difficulties encountered in obtaining amplification of the whole VP1 gene for sequencing suggested that the genomic viral RNA could be at least partially degraded and of poor quantity.

There are some, small-to-moderate, English language corrections to be made throughout the manuscript.

Answer: indeed, we realized it, the manuscript has now been completely revised

Title

Lines 2-3. The title is misleading. The FTA cards are a tool for rapid field sampling. Detection and serotyping in this article took place in a laboratory. Please provide alternative title.

Answer: thanks for pointing this out, we have provided a new title.

Introduction

Lines 25-28. The sampling is not clearly described in the Abstract section. Please rewrite.

Done.

Line 70. “…avoiding disease contamination.” Did you mean avoiding accidental disease transmission?

Answer: thank you, we have corrected the misleading sentence.

Lines 64-77. Please describe the background of the previous usage of FTA cards in FMDV related diagnostics. The general description of their characteristics for infectious diseases should be replaced by more focused data on FTA cards related to samples originated fromFMDV-infectedd animals and different sample types.

Answer: we have integrated the introduction accordingly

Materials and methods

Section 2.1-2.3 please provide the actual number of each sample type collected and refer to the table 1 for further description.

It is not clear whether these 16 samples originate from 16 animals or different sample types were collected from some individuals? Please provide clear description.

Answer: We have tried to improve the description of the samples as suggested 

Table 1. What does “-“ mean?

Answer: lesions in sheep are quite difficult to see and evaluate, thus it was not possible to age them. We have changed with N/A (not available), we also gave this explanation in the text.

Figure 1. Since FTA cards have been in use for some time, this figure seems unnecessary, or, you can provide it in the supplementary material.

Answer: we agree with the reviewer; the figure has been eliminated.

Line 127. Reaction conditions were not provided. If citation Callahan et al. covers it, please rewrite the sentence in lines 126-128.

Answer: we corrected it accordingly.

Line 128. Please explain the annealing/elongation temperature of 50.3°C.

Answer: thank you, it was a mistake, we corrected it.

Line 129. Primers not primes. Please correct. Add “-“ between real and time (the same error needs to be corrected throughout the manuscript).

Answer: we corrected it accordingly.

What did you mean by profile?

Answer: it was an oversight, thank you to point it out.

Table 2. Please provide positions of designed oligos in relation to the reference FMDV genome and which target gene fragment they amplify.

Answer: we have included the oligos position and the reference sequence as requested.

Please remove Table 3 and provide short description of reaction conditions within the text or provide citation only. Furthermore, the citation you have provided (Knowles et al., 2014) is not related to primers described in Table 3. Have you used the Knowles et al. from 2016 (VP1 sequencing protocol for foot and mouth disease virus molecular epidemiology)?

Answer: we have removed the table, the reference is Knowles et al. 2016 and it is reported in the bibliography.

Line 142. Semi-nested, please correct.

Done

Lines 151-153. The authors have not provided a necessary information regarding the multiple sequence alignment (method) and the phylogenetic analysis (methods, models, bootstrap) they implemented.

Answer: we integrated the text with all the information required

Line 153. If you provide the information that the Pirbright Institute is WRL, then provide the information that Anses/Sciensano are the EURL for FMD.

Answer: we rephrase the paragraph, only the sequences of the serotype O prototype have been downloaded from the WRL site, all the other sequences were present in Genbank.

It is not clear whether these sequences were deposited in GenBank; accession numbers were not provided.

Answer: the phylogenetic tree has been redone, new sequences have been included from Genbank and those produced in the framework of this study have been deposited in the GenBank database, the accession numbers are reported in the text and in the table.

Results

Line 159. Please correct the typos.

Done

Table 4. Please add an additional column for VP1 RT-PCR results.

Answer: the last column actually reports the results of the VP1 real-time RT-PCR. The misunderstanding could have been generated by the incongruency in the table 4 title. We corrected the title.

Lines 165-166. Figure 2 does not provide VP1 sequencing information.

Answer: we have rephrased it

Figure 2 is not informative and should be removed.

Answer: we have eliminated it

Line 170. The sentence is repeated since almost the same one was written in the line 156.

Answer: we have rephrased.

Lines 172-174. To claim these samples were negative, the results of IPC amplification should be provided. No information regarding IPC was provided in materials and methods. Please introduce these results and related information on the materials and methods used.

Answer: as at the FMD OIE/World Reference Laboratory at The Pirbright Institute, the real-time RT PCR test accredited according to ISO 17025 at IZSLER Reference Laboratory does not involve the use of IPC; however, test validation and implementation of the test had previously verified its applicability to the various possible matrices.

Lines 176-181. Have you used the FMDV Toolbox the additionally check the VP1 sequences?

Answer: We have described in the text the analysis of the sequences, in the phylogenetic analysis we have included the sequences of prototypes for serotype O which have been retrieved from the WRL for FMD.

Figure 3. The resolution is poor, please provide better Figure. Moreover, the accession numbers of all taxa in the phylogenetic tree should be provided within the tree, or as a supplementary material. Some taxa have asterisks in their name, please provide an explanation on the meaning. The overall number of taxa could be reduced in order to improve on clarity of that tree.

Answer: we corrected as suggested, we did not reduce the number of taxa but we hope the tree, that is been redone, is now improved.

Discussion

Lines 206-208. The explanation should be expanded on the comparisons with other studies using FTA cards related to FMDV detection. What is the limit of detection regarding FMDV load on FTA cards?

Answer: the discussion, as well as the whole manuscript, has been almost completely revised and we have provided further details in this regard.

Author Response

I’d like to thank all reviewers for their exhaustive revision, valuable, generous and constructive comments and suggestions, that for sure have great impact to improve our submitted manuscript.

Just to bring the attention to the reviewers and editors that this work has not been done in normal country, to work in a country like Libya, unstable country with ongoing war and conflict for few years, is not an easy task.  

I hope that all corrections made will satisfy the reviewers and Editors expectations and to have our manuscript published in the esteemed journal “Animals”!

Reviewer 3

The purpose of this article is trying to clarify “Using FTA cards as a rapid and field tool for detection and serotyping of foot-and-mouth disease virus in Libya”. However, there are not enough proofs in your article to convince readers. In addition, the contents, methods, and data analysis are too simple to meet the publication requirement in the journal.

Answer: we have extensively revised the manuscript trying to improve it and discuss our results in relation to previous reports of experimental studies using FTA cards and to the main objective of our report, which was simply the description of the experience of using FTA cards in a real case. We think that the answer given to the initial comment of reviewer n.2 may be pertinent also here.

  1. If you want to prove FTA cards can work as an alternative tool for FMDV samples’ transportation, you should include another regular method (cold-chain transportation) to do comparison. Only after doing different comparisons, the conclusion can be drawn which one is better or can work as an alternative tool.

Answer: no doubts that regular transportation of original samples is the optimal method to preserve the virus, including virus infectivity; however, this is not possible in several cases and easier alternative transportation systems are desired. Libya is unstable country with war and conflict for years, we could not find and courier is willing to ship clinical samples, hence we have used the FTA cards!

  1. As for the number of samples, more samples should be included. Only 16 samples from 2018-2019 were taken here. How about others after 2019?

Answer: Last FMD outbreak in Libya was in 2019. The National Center for Animal Health has started vaccination campaigns against FMD all over the country by vaccinating cattle twice per year and using ring vaccination for small ruminants.

  1. As for the efficiency of FAT transportation for nucleic acid’s test, only epithelial tissue ( with No. 16 missing) showed positive results. What is the deep reason behind it? It means oral swab or whole blood samples are not necessary to be collected later. It will lead to drawing a wrong conclusion.

Answer: the other referees pointed out this issue as well, and we have extensively revised the discussion integrating it with explanations.

  1. In method section 2.6: the second Real-Time RT-PCR was a serotype specific assay designed by IZSLER to detect isolates belonging to FMD virus O/EA-3 topotype. I wonder if this method can specifically detect FMD virus O/EA-3 topotype and what is the reference?

Answer: we have designed the primers and the probe considering the sequences of the strains circulating in Algeria in the same period to specifically detect EA/3 topotype.

  1. Figure 2 is not necessary.

Answer: we have eliminated it

  1. There are several inaccurate grammar usages,for example:

Sentences 164-166: “ Therefore, these positive seven positive samples for FMD virus serotype O and most likely to be topotype O/EA-3, that has been confirmed by VP1 gene sequencing (Figure 2). ”

Answer: the manuscript has now been completely revised

  1. Also some wrong typo there in the article.

Answer: the manuscript has now been completely revised

Round 2

Reviewer 2 Report

The manuscript has been improved, however, there is one major and some minor points that should be resolved before the manuscript can be considered for publication. There are still some minor errors in the English language. The present manuscript is rather a short communication or a case study than a complete article.

Major remark:

Authors: as at the FMD OIE/World Reference Laboratory at The Pirbright Institute, the real-time RT PCR test accredited according to ISO 17025 at IZSLER Reference Laboratory does not involve the use of IPC; however, test validation and implementation of the test had previously verified its applicability to the various possible matrices.

I am aware that IPC is not sometimes included under the 17025 ISO standard. However, this is a scientific report and it should contain the data on IPC since you cannot clearly state these negatives are true negatives. The control of possible inhibition in PCR is of utmost priority. There are very easy and convenient ways to detect the IPC, whether it is endogenous or exogenous. The authors should report these data.

Minor remarks:

Line 129. What does the normal courier mean? I suppose that the authors are referring to couriers without appropriate permit for the transport of biologically dangerous material. Please provide better wording for “normal”.

Table 1. For the vaccination, the authors should provide a clearer description in the footnotes, that vaccination status refers to the FMD.

Line 163. The provided reference is not correct. Please provide the reference which describes the actual procedure for the VP1 genotyping as previously suggested.

Line 182. The MEGA XI should be written as MEGA11. Moreover, the wrong reference was provided. Please update it to the correct reference for MEGA11 (Tamura et al., 2021).

Lines 178-187. The authors should provide the information regarding the number of sequenced strains and the GenBank accession number in the materials and methods section. These data are provided in results, section 3.2. Please rewrite.

Line 261. The authors are probably referring to 2018, please correct.

Lines 264-265. The sentence is not clear, please rewrite.

In discussion, the authors should provide an explanation for the VP1 amplification failure in 6 out of 7 samples. They have given possible causes in their answer to the first issue of the initial revision.

Line 289. The reference No. 35 seems to be incorrect since it is related to Egypt. Libya is not mentioned within that paper.

The conclusion seems too long. The introductory part is not necessary. The authors should summarize their findings as short as possible. Other information should be included in the discussion.

Author Response

The manuscript has been improved, however, there is one major and some minor points that should be resolved before the manuscript can be considered for publication. There are still some minor errors in the English language. The present manuscript is rather a short communication or a case study than a complete article.

Answer: Thank you so very much for time and excellent feedback and comments. We have done our best to satisfied with our answers to your valuable comments. Regarding your concern for the manuscript to be as a short communication or a case study; we can see that the other two reviewers were happy for the manuscript to be as an article, and all authors are keen to see it published as an article, as it is presenting a good piece of work that first time done in Libya and Italy and also the results of this work was very important for veterinary authority in Libya and the decision on FMD vaccine type introduced and used in Libya based mainly on the results of this work! 

Major remark:

Authors: as at the FMD OIE/World Reference Laboratory at The Pirbright Institute, the real-time RT PCR test accredited according to ISO 17025 at IZSLER Reference Laboratory does not involve the use of IPC; however, test validation and implementation of the test had previously verified its applicability to the various possible matrices.

I am aware that IPC is not sometimes included under the 17025 ISO standard. However, this is a scientific report and it should contain the data on IPC since you cannot clearly state these negatives are true negatives. The control of possible inhibition in PCR is of utmost priority. There are very easy and convenient ways to detect the IPC, whether it is endogenous or exogenous. The authors should report these data.

Answer: We have specified the inclusion in the 3D reaction of an exogenous synthetic RNA in the reaction master mix to evaluate the absence of PCR inhibitors. Though we have retested the samples, with the exogenous control working as expected in all samples, we did not modify the data originally obtained and presented in the table in order to not introduce possible bias due to the longer storage of the samples. This information was included in the revised manuscript. The internal control is a synthetic RNA was produced in IZSLER Lab in Italy, its production requires biomolecular techniques skills, including cloning procedure and in vitro transcription. For this reason, we have added a new author "Stefano Baselli" who has produced this control.

Minor remarks:

Line 129. What does the normal courier mean? I suppose that the authors are referring to couriers without appropriate permit for the transport of biologically dangerous material. Please provide better wording for “normal”.

Answer: by normal currier means inside plastic sealed bags in envelope by normal airmail post, without using the three-layer package, and in room temperature without the need for ice blocks or dry ice.

Table 1. For the vaccination, the authors should provide a clearer description in the footnotes, that vaccination status refers to the FMD.

Answer: Done, we have specified “FMD vaccinated” in the header of the relevant column of the table.

Line 163. The provided reference is not correct. Please provide the reference which describes the actual procedure for the VP1 genotyping as previously suggested.

Answer: We are sorry for the inaccuracy; we have added the new following accurate reference: Knowles NJ, Wadsworth J, Bachanek-Bankowska K, King DP. VP1 sequencing protocol for foot and mouth disease virus molecular epidemiology. Rev Sci Tech. 2016 Dec;35(3):741-755. doi:10.20506/rst.35.3.2565. 

Line 182. The MEGA XI should be written as MEGA11. Moreover, the wrong reference was provided. Please update it to the correct reference for MEGA11 (Tamura et al., 2021).

Answer: Accordingly, we have updated the reference with the one you suggested: Koichiro Tamura, Glen Stecher, Sudhir Kumar. MEGA11: Molecular Evolutionary Genetics Analysis Version 11. Mol Biol Evol. 2021 Jun 25;38(7):3022-3027. doi: 10.1093/molbev/msab120.

Lines 178-187. The authors should provide the information regarding the number of sequenced strains and the GenBank accession number in the materials and methods section. These data are provided in results, section 3.2. Please rewrite.

Answer: We have modified the required information in the revised manuscript as the reviewer suggested.

Line 261. The authors are probably referring to 2018, please correct.

Answer: You're right. Our mistake and has been corrected.

Lines 264-265. The sentence is not clear, please rewrite.

Answer: Thank you for the advice, we have rephrased the sentence to be more readable and understandable

In discussion, the authors should provide an explanation for the VP1 amplification failure in 6 out of 7 samples. They have given possible causes in their answer to the first issue of the initial revision.

Answer: Thank you, we have given more explanation as requested

Line 289. The reference No. 35 seems to be incorrect since it is related to Egypt. Libya is not mentioned within that paper.

Answer: The reference has been removed. 

The conclusion seems too long. The introductory part is not necessary. The authors should summarize their findings as short as possible. Other information should be included in the discussion.

Answer: Agreed and the last paragraph of the conclusion has been moved as a last part of the discussion section.

Thank you again with best wishes

Round 3

Reviewer 2 Report

The authors have improved the manuscript. However, there are still some minor remarks to be corrected.

Lines 165 and 278. Please specify that RT-PCR was real-time RT-PCR.

Line 193. The name of the strain (O/LIB/Misrata/2019) should be written earlier in the sentence.

The conclusion was not shortened as suggested by authors. As I have previously said, the conclusion should be as short as possible. The last paragraph is not the only problem (not removed in the revised manuscript); the conclusion even has introductory part which should be removed.

Lines 417-423. The previous incorrect reference was not deleted. Please delete it.

Line 465. The reference starts with an error.

There are still some minor English language errors to be corrected.

Author Response

Dear Reviewer,

Thank you so very much for your valuable comments. However, I have to clarify something important, as all comments raised by you in the first and second rounds were addressed and submitted in our revised manuscript as Word File with Tack Changes. Whereas all your new comments were based on the revised PDF File, which is the old revised manuscript not the new revised one! The PDF file of the manuscript does not match with the word file.

Lines 165 and 278. Please specify that RT-PCR was real-time RT-PCR.

Answer: Done!

Line 193. The name of the strain (O/LIB/Misrata/2019) should be written earlier in the sentence.

Answer: Done!

The conclusion was not shortened as suggested by authors. As I have previously said, the conclusion should be as short as possible. The last paragraph is not the only problem (not removed in the revised manuscript); the conclusion even has introductory part which should be removed.

Answer: You have accessed the PDF file, the old revised version. In our revised word file manuscript, the Conclusion section was reduced as suggested by the reviewer.

Lines 417-423. The previous incorrect reference was not deleted. Please delete it.

Answer: As above, the PDF is not the same as the revised Word file. This was removed as well from the manuscript word file as suggested by the reviewer.

Line 465. The reference starts with an error.

Answer: As above. I even do not have line with the number 465 in the word file. This was the old PDF file.